# Mental health disorders, and associated factors among children aged 6–17 years living in Mahama refugee camp in Rwanda

Allan Higiro[1,2]*, Dan Lutasingwa[1], Mickel-Ange Karamage[1], Everest Turatsinze[1], Ritah Mukashyaka[1], Deborah Kansiime[1], Odile Habimana[1], Angelique Uwigiciro[1], Elisabeth Iriza[1], Norbert Tuyishimire[1], Fred Mulisa[1], Egide Niyotwagira[1], Moses Ochora[3], Daniel Gatei Waweru[4], Aflodis Kagaba[1], Alain Favina[1,5]

1 Research Department, Health Development Initiative, Kigali, Rwanda, 2 Department of Paediatrics, African Health Sciences University, Kigali, Rwanda, 3 Department of Paediatrics, Faculty of Medicine, Soroti University, Soroti, Uganda, 4 Save the Children, Kigali, Rwanda, 5 Department of Psychiatry, Faculty of Medicine, Mbarara University of Science and Technology, Mbarara, Uganda

* allanhigiro1@gmail.com

## Abstract

Children living in refugee camps are negatively affected by adverse mental health outcomes due to traumatic experiences and living conditions such as food insecurity. However, the mental health disorders of children living in refugee camps of Rwanda have not yet been empirically reported. This study, therefore, assessed the prevalence and associated factors of depression, post-traumatic stress disorder (PTSD), and suicidal ideation among children living in a refugee setting, with particular focus on the association with food insecurity. This was a cross-sectional study conducted among 500 children aged 6–17 years living in Mahama Refugee Camp, Rwanda. The MINI-KID tool was used to assess for mental health disorders and the Household Food Insecurity Access Scale (HFIAS) for food insecurity. Descriptive, bivariate, and multivariate logistic regression analyses were performed using STATA-17. The prevalence of depression, PTSD, and suicidal ideation were 12.6%, 3.0% and 2.2% of participants, respectively. Severe food insecurity was prevalent at 69.4% (347/500). Food insecurity was only associated with depression (aOR=1.12, 95%CI: 1.07-1.17, p<0.001). Additional factors significantly associated to depression were age (aOR=1.26, 95%CI: 1.13-1.42, p<0.001), and coming from DRC (aOR=2.83, 95%CI: 1.38-5.79, p=0.005). PTSD was associated with coming from Democratic Republic of Congo (DRC) (aOR=4.83, 95%CI: 1.55-15.04, p=0.007), while suicidal ideation was also associated with coming from DRC (aOR=6.91, 95%CI: 1.79-26.69, p=0.003) and having a disability (aOR=9.44, 95%CI: 1.83-48.77, p=0.007). The prevalence of mental health disorders among refugee children in Mahama Camp are high with depression being significantly associated with food insecurity. Integrative interventions addressing both mental health and food insecurity among children like encouraging modern agricultural practices in humanitarian settings are crucial. Supporting

**Data availability statement:** All data generated and analyzed during this study are presented in the manuscript.

**Funding:** Save the Children–Rwanda provided funding for data collection via its project on strengthening community health work in Mahama refugee camp. The grant number associated with the funding is 18818. The funding was received by the Health Development Initiative, a local non-government organization, and no author in particular received or used this funding. The funders had no role in the study design, data collection and analysis, the decision to publish, or the preparation of the manuscript. The content is solely the responsibility of the authors.

**Competing interests:** The authors have declared that no competing interests exist.

children living with disability according to their individual needs can further enhance their mental health.

---

## 1. Introduction

The global refugee crisis has resulted in high levels of forced displacement with children constituting a significant proportion of affected populations. As of 2024, the United Nations High Commissioner for refugees (UNHCR) estimates that children aged 0–17 years account for 40% among refugees [1]. Rwanda hosts a significant refugee population, with Mahama Refugee Camp in Kirehe district being the largest camp in the country [2]. Established in 2015, Mahama refugee camp hosts over 54,000 refugees, more than half of whom are children under 18 years of age [3]. Children in humanitarian settings are particularly vulnerable group of people due to their ongoing childhood developmental needs, sensitivity to traumatic experiences, and dependence on consistent family and community support [4]. When these needs are not met, this may lead to severe consequences, including mental health disorders [5]. Previous studies conducted in other countries have highlighted the increasing prevalence of mental health disorders among refugee children. Depressive symptoms were reported in 41% and PTSD in 36% of Syrian refugee children in Lebanon [6], while in Jordan, 27% had suicidal ideation [7]. A narrative review conducted in 2019 revealed that about 80% of children refugees aged 8 years and below have mental health challenges including depression and PTSD [8] while another study revealed 5.3% of Sudanese refugee adolescents in Uganda experienced suicidal ideation [9]. However, data on the mental health of refugee children in Rwanda remains scarce.

Food insecurity has emerged as a critical social determinant of child mental health. A systematic review conducted among non-refugee children in the US revealed that there was a significant association between food insecurity and mental health outcomes including depression and anxiety [10]. Similarly, in Canada, children aged 5–11 years from food insecure households had higher odds of diagnosed mental health conditions [11]. In Sub-Saharan Africa, data from Global School-Based Student Health Survey (GSHS) among children aged 13–17 years from various African countries including Tanzania and Uganda showed that food insecurity was associated with increased prevalence ratios for suicidal planning [12].

Food insecurity has emerged as a critical social determinant of child mental health, operating through pathways such as chronic stress, nutritional deprivation, and household instability. A systematic review conducted among non-refugee children in the United States demonstrated significant associations between food insecurity and mental health outcomes, including depression and anxiety [10]. Similarly, a population-based study in Canada reported higher odds of diagnosed mental health conditions among children aged 5–11 years living in food-insecure households [11]. In sub-Saharan Africa, analyses of Global School-Based Student Health Survey (GSHS) data among adolescents aged 13–17 years identified food insecurity as a significant correlate of suicidal behaviors, including suicidal planning [12]. These

findings highlight a consistent association between food insecurity and adverse mental health outcomes across diverse child populations.

The relationship between food insecurity and mental health is crucial to explore among children refugees, as inadequate nutrition not only affects physical health but also exacerbates psychological distress [13]. Despite the known relationship between food insecurity and mental health disorders in other populations, little is known about this relationship among refugee children. Several studies have mainly focused on mental health of children in refugee settings, or the association of food insecurities and mental health of children in non-refugee setting. In addition, the mental health condition of children in Rwanda have not yet been empirically documented. This study therefore aimed to explore prevalence of and the association between depression, PTSD and suicidal ideation, food insecurity and other associated factors among children living in Mahama Refugee Camp. The findings can help inform policies and interventions that support both mental health and nutritional needs of refugee children in Rwanda and beyond.

Among refugee children, the relationship between food insecurity and mental health warrants particular attention. Inadequate and uncertain access to food not only compromises physical health but may also exacerbate psychological distress, especially in contexts marked by displacement and trauma [13]. However, existing studies have largely examined either mental health outcomes in refugee children or the association between food insecurity and mental health in non-refugee populations. Furthermore, the mental health status of refugee children in Rwanda has not been empirically documented. This study therefore aimed to assess the prevalence of depression, PTSD, and suicidal ideation, and to examine their associations with food insecurity and other sociodemographic factors among children aged 6–17 years living in Mahama Refugee Camp, Rwanda.

## 2. Methods

### 2.1. Study setting and design

This was a cross-sectional study that was conducted in Mahama refugee camp, the largest refugee camp in Rwanda, located in Kirehe district, in the Eastern province.

### 2.2. Study population, inclusion and exclusion criteria

This study involved children living in Mahama refugee camp. Children were eligible for inclusion if they were aged between 6–17 years and excluded if they had lived in the camp for less than 6 months.

### 2.3. Sampling technique

Participants were selected using a convenience sampling technique which allowed us to efficiently recruit willing participants who met the eligibility criteria.

### 2.4. Sample size

The sample size was calculated using the Cochran's formula for estimating a population proportion in cross-sectional studies. We used an estimated prevalence of 50%, a 95% confidence level, and a 5% margin of error, which yielded a minimum required sample of n = 384 participants. After adjusting for a non-response rate, the final target sample size was n = 500 participants

### 2.5. Data collection procedure

After receiving ethical approval, a team of ten trained and experienced research assistants collected data. They first met with the camp leaders and village community health worker, who helped introduce the study to the community. Together, they moved through the community announcing the research project and inviting families to participate. Each village was

assigned a research assistant and community health worker, who then visited homes explaining the study objectives, procedures and benefits to both children and their parents or guardians. Families who agreed to participate were included, and to ensure fairness, one eligible child per home was allowed to participate. Parents or guardians provided written informed consent, while children gave their assent prior the interview. Data was collected from 12th November 2024–17th November 2024, using a structured data extraction form by research assistant who were fluent in Kinyarwanda and Swahili, the primary spoken languages in Mahama Refugee camp and they had received standardized training on the administration of all study instruments to ensure consistency.

## 2.6. Measurement tools

The data extraction form included questions on socio-demographic characteristics, mental health and food security. The socio-demographic characteristics included the date of birth, gender, education status, class of study, orphan status, nationality and disability status.

The mental health assessments were conducted using the Mini-kid tool. This is a structured instrument designed to assess mental health disorders in children aged 6–17 years based on DSM-5 (Diagnostic and Statistical Manual of Mental health, fifth edition) and ICD-10 (International Classification of Disease, tenth edition) [14]. This tool has been used in some studies done among children in Sub-Saharan Africa including Rwanda and was proven to be valid [15]. We assessed depression, PTSD and suicidal ideation which were assessed using a set of questions in the Mini-kid tool, analyzed and categorized as present or absent (yes/no) as per the scoring criteria in the Mini-kid tool [16]. This tool has been used before to assess mental health among children in other refugee settings [17,18].

Household Food Insecurity Access Scale (HFIAS) was used to assess severity of food insecurity experiences in the last 30 days. The HFIAS consists of nine questions where each questions had a Likert scale of never, rarely, sometimes and often [19]. Depending on how specific questions are answered in the tool, participants become categorized into 4 different categories; food secure, mildly food insecure, moderately food insecure and severely food insecure [20]. Households in the food secure category reported no food insecurity experiences, while those classified as mild food insecurity indicated occasional or mild anxiety about food shortages without significant impact on food access. Moderate food insecurity was identified in households that experienced more frequent reductions in food quality or quantity, or resorted to coping strategies such as reducing meal sizes or skipping meals while those in the Severe food insecurity category included households that faced extensive food shortages, leading to severe disruptions in eating habits, such as going without meals for extended periods [21]. In this study, the Cronbach's alpha was 0.96.

The HFIAS was scored following standard procedures, generating both a continuous total score (range 0–27, with higher scores indicating greater food insecurity) and categorical classifications (food secure, mildly food insecure, moderately food insecure, and severely food insecure). In this study, the categorical HFIAS classification was used to describe the distribution of food insecurity levels in the sample (S1 Fig), while the continuous HFIAS score was used in bivariate and multivariate regression analyses to assess associations with mental health outcomes. This approach was chosen to preserve statistical power.

## 2.7. Data analysis

We analyzed the data using STATA-17. Descriptive statistics were presented after checking for the Gaussian assumption. The results were reported as proportions, percentages for categorical variables whereas median and interquartile range for the continuous variable; age. A comparison of demographic between people with or without a mental health concern were performed using Chi-square and fisher exact for the categorical predictors, Mann-Whitney U test for non-parametric continuous variable predictor (age). Bivariate and multivariate logistic analyses were done to examine associations of different predictor to the outcomes. The multivariate analysis helped to adjust for potential confounders, and got adjusted

odds ratios (aORs) and p-values after testing for collinearity where the variance inflation factor (VIF) was below two. A p-value of less than or equal to 0.05 was considered significant for a 95% confidence interval.

### 2.8. Ethics statement

The study was conducted in accordance with the principles of the Declaration of Helsinki [22]. Ethical approval was obtained from the Rwanda national ethics committee (RNEC; IRB 00001497) and Ministry in Charge of Emergency Management (MINEMA). Administrative clearance was granted by the Kirehe District authorities and Mahama Refugee Camp leadership. Written informed consent was obtained from the children's parents/guardians and assent from the participating children. Participation was voluntary, and participants were informed of their right to withdraw at any time without consequences. Data were collected anonymously to protect participant confidentiality.

## 3. Results

### 3.1. Study participants

In total, the study involved 500 children aged between 6–17 years in Mahama refugee camp. Most of them were aged between 10–14 years (284/500; 56.8%). Slightly more than a half of participants (278/500; 55.6%) were female, most of the participants were Burundian (425/500; 85%) and 348 (69.4%) were severely food insecure (S1 Table).

### 3.2. Distribution of sociodemographic factors and food security across depression, PTSD and suicidal ideation

The prevalence of depression, PTSD, and suicidal ideation was 12.6% (63/500), 3.0% (15/500), and 2.2% (11/500) respectively. Depression was more common among older children ($p < 0.001$), females ($p = 0.010$), Congolese ($p < 0.001$), those lacking at least one parent ($p = 0.004$), children with disabilities ($p = 0.031$), and those experiencing higher levels of food insecurity ($p < 0.001$). PTSD was significantly associated with older age ($p = 0.042$), Congolese nationality ($p < 0.001$), disability ($p = 0.031$), and food insecurity ($p = 0.020$). Suicidal ideation was more prevalent among older children ($p = 0.032$), females ($p = 0.017$), Congolese ($p < 0.001$), children with disabilities ($p < 0.001$), and those experiencing greater food insecurity ($p = 0.038$). (S1 Table).

### 3.3. Distribution of households by food insecurity based on HFIAS classification

Among the 500 children, 22.4% lived in food secure households, 3.2% in mildly food insecure households, 5.0% in moderately food insecure households and 69.4% in severely food insecure (S1 Fig).

### 3.4. Factors associated with depression, PTSD, suicidal ideation

Bivariate (S3 Table) and multivariate (S2 Table) logistic regression were done.

The results from the multivariate logistic regression showed that food insecurity was associated with depression (aOR=1.12, 95%CI: 1.07-1.17, $p < 0.001$). Other factors associated with depression were age (aOR=1.26 95% CI: 1.13-1.42, $p < 0.001$), coming from DRC (aOR=2.83, 95%CI: 1.38-5.79, $p = 0.005$). Food insecurity was not associated with PTSD (aOR=1.07, 95%CI: 0.99-1.15, $p = 0.096$). The factor significantly associated with PTSD was coming from DRC (aOR=4.83, 95%CI: 1.55-15.04, $p = 0.007$). Food insecurity was not associated with suicidal ideation (aOR=1.05, 95%CI: 0.96-1.15, $p = 0.275$). The factors significantly associated with suicidal ideation was coming from DRC (aOR=6.91, 95%CI: 1.79-26.69, $p = 0.003$) and having disability (aOR=9.44, 95%CI: 1.83-43.77, $p = 0.007$).

## 4. Discussion

This study explored the prevalence and associated factors of depression, post-traumatic stress disorder (PTSD), and suicidal ideation among children aged 6–17 years living in Mahama refugee camp in Rwanda with particular focus on the

association with food insecurity. The findings showed high prevalence of depression (12.6%), PTSD (3.0%), and suicidal ideation (2.2%). A significant proportion (69.4%) of the children lived in households experiencing severe food insecurity and food insecurity was only associated with depression. Other factors associated with depression was age. Being a refugee from DRC was associated with depression, PTSD and suicidal ideation. Having a disability was only associated with suicidal ideation.

The mental health challenges observed in this study are consistent with previous research showing elevated mental health concerns among displaced children and adolescents [6–9]. A 2021 multilevel review found that refugee children frequently experience depression ranging from 10% to 62% depending on the trauma history and the conditions in their host environment [23]. The prevalence of PTSD in this study was comparable to the 4.7% prevalence of PTSD found among refugee children in a study conducted in Germany [24]. This similarity highlights that PTSD is influenced not only by pre-migration trauma but also by post-migration stressors [25]. Refugees in high-income countries may face new challenges like discrimination and asylum uncertainty that may sustain PTSD symptoms [26], while children in camps may benefit from community cohesion that buffers distress [27]. These findings highlight the complex interplay of trauma, environment, and resilience across diverse refugee contexts.

Additionally, the prevalence of suicidal ideation in this study lies in the range reported in the systematic that reported prevalence ranging between 0.17% to 70.6% [28]. The relatively lower prevalence of suicidal ideation may be due to fact that Congolese children, who had significantly high prevalence of PTSD and suicidal ideation, made up only 15% of the study population. Nonetheless, even low prevalence in childhood is alarming considering the lifelong consequences if left unaddressed.

This study found a positive association between food insecurity and depression. These results are similar to previous findings in a study done among children aged 5–11years in Canada that reported that children from severely food-insecure households had 1.67 times higher odds of being diagnosed with a mental health conditions than their food-secure peers [11]. Other studies have linked food insecurity not only to emotional problems but also to behavioral challenges among displaced populations [29,30]. Household food insecurity is associated with persistent stress and worry, which can dysregulate stress-response systems and contribute to depressive symptomatology [31]. Food insecurity may also disrupt family functioning and caregiver mental health, intensifying emotional strain within the household. Research indicates that parental stress and depression correlated with food insecurity can mediate adverse outcomes in children's development and psychological wellbeing, highlighting the importance of the family environment in the pathway from food insecurity to child mental health difficulties [32]. Either from refugee setting or not, food insecurity exacerbates daily stress, impairs concentration, and impacts emotional stability in children, compounding the impact of trauma [33].

The high levels of household food insecurity observed in this study likely reflect a combination of pre-displacement vulnerabilities and post-displacement structural constraints commonly experienced by refugee populations [34]. Many children living in Mahama Refugee Camp originate from regions affected by prolonged conflict and economic instability, which may have disrupted livelihoods, education, and access to food even prior to displacement. Studies among refugee populations have repeatedly shown that exposure to prolonged conflict and associated socioeconomic disruption contributes to sustained household vulnerability, including food insecurity, after displacement [35,36]. After arrival in camp settings, persistent food insecurity may be sustained by limitations in food assistance and livelihood opportunities. Refugee settings often involve reliance on external food aid that may be insufficient to meet household needs, seasonal shortages in food distributions, and restricted access to labor markets or agricultural land, all of which are associated with chronic food insecurity among displaced households [37,38]. Food assistance programs implemented by humanitarian agencies aim to reduce acute hunger, but they may not fully address long-term food adequacy and dietary diversity [39,40].

In this study, older children had increased odds of depression, which may reflect cumulative exposure to traumatic events or increased cognitive awareness of hardships. This finding was also observed among older Syrian refugee adolescents in Turkey [41]. Disability was significantly associated with suicidal ideation. A study done in the US reported that

children living with disability had three times higher odds of exhibiting suicidal behaviors [42]. This vulnerability may arise from social exclusion, stigma, experiences of bullying, and difficulties accessing supportive mental health services which compound feelings of hopelessness and distress [43].

Being of Congolese nationality was associated with increased odds of depression, PTSD, and suicidal ideation. Although not directly explored in this study, this may reflect underlying psychosocial vulnerabilities such as recent exposure to violence, prolonged displacement, and social marginalization that have been shown to significantly affect the mental wellbeing of Congolese refugees in similar settings, including Rwanda and Uganda [44]. Social marginalization of Congolese refugees in Mahama refugee camp might be explained by their smaller percentage and recent integration into this camp [45]. Recent studies conducted among Congolese refugee populations in East Africa have documented high levels of cumulative trauma exposure, including witnessing violence, family separation, and insecurity during migration, all of which are strongly associated with depressive and post-traumatic stress symptoms among children and adolescents [46]. Additionally, refugee youth who experience minority status within camps or host communities may face heightened social exclusion, reduced peer integration, and perceived discrimination, factors that independently contribute to internalizing symptoms and suicidal ideation [47]. In contexts where Congolese refugees represent a smaller and more recently integrated subgroup like in Mahama refugee camp, these structural and interpersonal stressors may amplify pre-existing trauma-related vulnerabilities hence may help explain the consistently higher odds of depression, PTSD, and suicidal ideation observed among Congolese children in this study.

Strengths and limitations: This study contributes to a growing but still limited body of evidence on refugee child mental health in sub-Saharan Africa. It is among the few that provide quantitative analysis that show the association between food insecurity and mental health concerns among children in refugee setting using validated tools. However, its cross-sectional design limits causal inference. The use of convenience sampling is also a limitation due to the likelihood of bias hence limiting generalizability. Additionally, the HFIAS tool, though validated, were mainly design to be responded one person in the household who could be usually the head. However, the tool had an excellent internal consistency as the questions from the tool are related to the experiences or frequency of reductions in food quantity, such as reducing meal sizes or skipping meals, facing extensive food shortages, or going without meals for extended periods. Additionally, it has been used among adolescents in other studies [48,49] supporting its applicability in this population.

## 5. Conclusion

This study highlights the mental health burden among children in Mahama refugee camp, with food security as a key associated factor for depression. The results call for efforts in reducing barriers to basic needs such as improving their food security through promoting modern agricultural practices within refugee settings [50]. Supporting children living with disability to feel included and valued according to their individual needs can further enhance their integration and overall wellbeing. While these findings highlight the potential relevance of food insecurity as a contextual factor linked to child mental health in humanitarian settings, the cross-sectional design of this study limits conclusions regarding causality or directionality. Nevertheless, the findings suggest that mental health screening and psychosocial support for refugee children may benefit from consideration of household food insecurity within broader social protection frameworks. Longitudinal and intervention studies are needed to clarify temporal pathways and to determine whether improvements in food security are associated with subsequent improvements in mental health outcomes.

## Supporting information

**S1 Table. Distribution of sociodemographic factors and Food security across Depression, PTSD and suicidal ideation.**
(DOCX)

**S1 Fig. Distribution of households by food insecurity based on HFIAS classification.**
(DOCX)

**S2 Table. Factors associated with Depression, PTSD, suicidal ideation.**
(DOCX)

**S3 Table. Factors associated with Depression, PTSD, suicidal ideation.**
(DOCX)

**S1 Data. Raw data used to generate the results in the manuscript.**
(XLSX)

## Author contributions

**Conceptualization:** Dan Lutasingwa, Daniel Gatei Waweru, Aflodis Kagaba, Alain Favina.

**Data curation:** Allan Higiro, Egide Niyotwagira, Alain Favina.

**Formal analysis:** Allan Higiro, Alain Favina.

**Investigation:** Allan Higiro.

**Methodology:** Allan Higiro, Dan Lutasingwa, Daniel Gatei Waweru, Alain Favina.

**Project administration:** Daniel Gatei Waweru, Aflodis Kagaba.

**Supervision:** Dan Lutasingwa, Daniel Gatei Waweru, Alain Favina.

**Validation:** Allan Higiro, Dan Lutasingwa, Daniel Gatei Waweru, Aflodis Kagaba, Alain Favina.

**Visualization:** Allan Higiro, Dan Lutasingwa, Daniel Gatei Waweru, Alain Favina.

**Writing – original draft:** Allan Higiro, Dan Lutasingwa, Alain Favina.

**Writing – review & editing:** Allan Higiro, Dan Lutasingwa, Mickel-Ange Karamage, Everest Turatsinze, Ritah Mukashyaka, Deborah Kansiime, Odile Habimana, Angelique Uwigiciro, Elisabeth Iriza, Norbert Tuyishimire, Fred Mulisa, Egide Niyotwagira, Moses Ochora, Daniel Gatei Waweru, Aflodis Kagaba, Alain Favina.

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
