## [Decision Letter · Decision Letter 0]

25 Nov 2025

PMEN-D-25-00426

Mental health disorders, and associated factors among children aged 6-17 years living in Mahama refugee camp in Rwanda.

PLOS Mental Health

Dear Dr. Higiro,

Thank you for submitting your manuscript to PLOS Mental Health. After careful consideration, we feel that it has merit but does not fully meet PLOS Mental Health’s publication criteria as it currently stands. Therefore, we invite you to submit a revised version of the manuscript that addresses the points raised during the review process.

Please find detailed comments from two reviewers, and myself, on the manuscript. Please ensure that these comments are either addressed in the manuscript or responded to in a rebuttal letter which outlines the changes, if any, that are made to the manuscript on resubmission.

I would also advise paying specific attention to the reviewer's comments re: data availability. The journal does request that the data associated with the presented manuscript is made available and that a statement is provided in the manuscript which explains how the presented work complies with the journal's data availability policy (Data Availability | PLOS Mental Health). Please ensure that the availability of data relating to the study is addressed in your resubmission.). Please ensure that the availability of data relating to the study is addressed in your resubmission.). Please ensure that the availability of data relating to the study is addressed in your resubmission.). Please ensure that the availability of data relating to the study is addressed in your resubmission.

We look forward to receiving your revised manuscript.

Kind regards,

Robert C. Dempsey

Academic Editor

PLOS Mental Health

Journal Requirements:

1. We noticed you have some minor occurrence of overlapping text with the following previous publication(s), which needs to be addressed:

- https://doi.org/10.3389/fped.2023.1037238

- doi: 10.1002/nop2.70137

In your revision ensure you cite all your sources (including your own works), and quote or rephrase any duplicated text outside the methods section. Further consideration is dependent on these concerns being addressed.

i. Please clarify all sources of financial support for your study. List the grants, grant numbers, and organizations that funded your study, including funding received from your institution. Please note that suppliers of material support, including research materials, should be recognized in the Acknowledgements section rather than in the Financial Disclosure.

ii. State the initials, alongside each funding source, of each author to receive each grant. For example: "This work was supported by the National Institutes of Health (####### to AM; ###### to CJ) and the National Science Foundation (###### to AM)."

iii. State what role the funders took in the study. If the funders had no role in your study, please state: “The funders had no role in study design, data collection and analysis, decision to publish, or preparation of the manuscript.”

iv. If any authors received a salary from any of your funders, please state which authors and which funders.

3. Please ensure that your Ethics Statement is available in its entirety at the beginning of your Methods section, under a subheading 'Ethics Statement'.

4. Please upload separate figure files in .tif or .eps format. Also, remove the figures from your manuscript file but keep the legends.

https://journals.plos.org/mentalhealth/s/figures

https://journals.plos.org/mentalhealth/s/figures#loc-file-requirements

5. We notice that your supplementary tables are included in the manuscript file. Please remove them and upload them with the file type 'Supporting Information'. Please ensure that each Supporting Information file has a legend listed in the manuscript after the references list.

6. We have noticed that you have uploaded Supporting Information files, but you have not included a list of legends. Please add a full list of legends for your Supporting Information files after the references list.

7. We note that your Data Availability Statement is currently as follows: “All data generated and analyzed during this study are presented in the manuscript”

Additional Editor Comments (if provided):

This is a very interesting population and area of concern. The manuscript has some good writing in places and structure - however, there are number of improvements that could be made to improve the quality of the communication of your findings. I have added some thoughts below alongside the reviewer's comments.

Please can you include page and line numbers in a resubmission - this helps facilitate peer review.

Abstract - please use the full wording for DRC in the abstract on first mention, I presume this is referring to the Democratic Republic of the Congo?

Page 4: "A review done in 2019" could be better phrased - was this a systematic literature review or some other type of review (e.g., review of policy documents, medical records)? Please clarify. A similar point applies on the next page in reference to a 'review done' - a 'review was conducted' would be better wording.

Methods: was there any translation or adaptation of the study materials for the sample? Where there any challenges with literacy with the target sample given their status as refugees?

Results: "Most of them were aged between 10-14 years" - this could be clearer and more specific, i.e. how many were aged between 10-14 years? Could you provide a percentage or raw number here?

Food insecurity - it would be useful to have a bit more commentary in the manuscript about how/why the participants had high levels of food insecurity - was this primarily due to their experiences before arriving at the refugee camp and/or were there specific issues with food availability/security in the camp itself?

Table 1 - it could be much clearer what the specific analyses are that are reported in this table - it was not easy to understand what the p values were associated with in the Table or which outcome relates to which analysis.

There also seemed to be remarkably low incidences of PTSD and suicidal ideation in the sample, is this correct?

Figure 1 - this is also difficult to interpret due to the presentation of both the percentages and raw numbers for each group - is this bar graph necessary or could this information be better presented in the main text?

As a non-African researcher based in Europe, I found it difficult to unpack why the 'DRC' was a key factor in the regression models - I am presuming that it is less about refugees coming from this specific country and more about instability and possible trauma/fighting in the DRC that is key? Some clarification on why coming from the DRC to this specific refugee camp is a key risk factor for poorer outcomes would be helpful.

"It is among the few that provide quantitative analysis that prove the association between food insecurity and mental health concerns among children in refugee setting using validated tools" - I would query how this has been 'proved' from a single study, I think this could be better phrased in the text to not suggest a definite finding/pattern.

Reviewers' comments:

Reviewer's Responses to Questions

**Comments to the Author**

1. Does this manuscript meet PLOS Mental Health’s publication criteria? Is the manuscript technically sound, and do the data support the conclusions? The manuscript must describe methodologically and ethically rigorous research with conclusions that are appropriately drawn based on the data presented.? Is the manuscript technically sound, and do the data support the conclusions? The manuscript must describe methodologically and ethically rigorous research with conclusions that are appropriately drawn based on the data presented.

Reviewer #1: Yes

Reviewer #2: Yes

2. Has the statistical analysis been performed appropriately and rigorously?

Reviewer #1: Yes

Reviewer #2: Yes

3. Have the authors made all data underlying the findings in their manuscript fully available (please refer to the Data Availability Statement at the start of the manuscript PDF file)?

The PLOS Data policy requires authors to make all data underlying the findings described in their manuscript fully available without restriction, with rare exception. The data should be provided as part of the manuscript or its supporting information, or deposited to a public repository. For example, in addition to summary statistics, the data points behind means, medians and variance measures should be available. If there are restrictions on publicly sharing data—e.g. participant privacy or use of data from a third party—those must be specified.requires authors to make all data underlying the findings described in their manuscript fully available without restriction, with rare exception. The data should be provided as part of the manuscript or its supporting information, or deposited to a public repository. For example, in addition to summary statistics, the data points behind means, medians and variance measures should be available. If there are restrictions on publicly sharing data—e.g. participant privacy or use of data from a third party—those must be specified.

Reviewer #1: Yes

Reviewer #2: No

4. Is the manuscript presented in an intelligible fashion and written in standard English?

Reviewer #1: Yes

Reviewer #2: Yes

Reviewer #1: 1. The Introduction provides good context but could benefit from tighter integration of the food insecurity–mental health link to strengthen the rationale.

2. The methods section is generally clear, but the choice of convenience sampling should be discussed more thoroughly, including its limitations for generalizability.

3. Please specify whether the MINI-KID tool was culturally adapted or validated in this refugee setting to strengthen credibility.

4. The calculation of sample size is well described, but consider simplifying the explanation to improve readability for non-statistical readers.

5. The Results section is comprehensive, but the large tables could be simplified or moved to supplementary materials for better flow.

6. The Discussion effectively relates findings to existing literature, but please reduce redundancy by avoiding repeating prevalence data multiple times.

7. Consider expanding on the potential mechanisms linking food insecurity and depression in this setting to strengthen interpretation.

8. The Conclusion is concise but could be more balanced by tempering strong policy recommendations with acknowledgment of the study’s design limitations.

9.Please check consistency in reporting prevalence (e.g., 69.4% vs 69.6% food insecurity) to avoid small discrepancies.

Reviewer #2: The manuscript addresses an important question and is methodologically appropriate, with conclusions supported by the data. Statistical analyses are appropriate. The authors did not provide access to the data used in the manuscript, which does not meet requirements and should be addressed. The manuscript is clear and intelligible. No concerns regarding dual publication or research ethics were identified. Overall, the study provides valuable insights and could be suitable for publication with major and minor revisions. A more detailed review of the manuscript has been uploaded.

**Do you want your identity to be public for this peer review?** For information about this choice, including consent withdrawal, please see our Privacy Policy..

Reviewer #1: **Yes:** Bala NimmanaBala NimmanaBala NimmanaBala Nimmana

Reviewer #2: No

---

## [Decision Letter · Decision Letter 1]

12 Feb 2026

Mental health disorders, and associated factors among children aged 6-17 years living in Mahama refugee camp in Rwanda.

PMEN-D-25-00426R1

Dear Dr Higiro,

We are pleased to inform you that your manuscript 'Mental health disorders, and associated factors among children aged 6-17 years living in Mahama refugee camp in Rwanda.' has been provisionally accepted for publication in PLOS Mental Health.

Best regards,

Robert C. Dempsey

Academic Editor

PLOS Mental Health

Reviewer Comments (if any, and for reference):

Reviewer's Responses to Questions

**Comments to the Author**

Reviewer #1: All comments have been addressed

Reviewer #2: All comments have been addressed

publication criteria? Is the manuscript technically sound, and do the data support the conclusions? The manuscript must describe methodologically and ethically rigorous research with conclusions that are appropriately drawn based on the data presented.? Is the manuscript technically sound, and do the data support the conclusions? The manuscript must describe methodologically and ethically rigorous research with conclusions that are appropriately drawn based on the data presented.

Reviewer #1: Yes

Reviewer #2: Yes

3. Has the statistical analysis been performed appropriately and rigorously?

Reviewer #1: Yes

Reviewer #2: Yes

4. Have the authors made all data underlying the findings in their manuscript fully available (please refer to the Data Availability Statement at the start of the manuscript PDF file)?

The PLOS Data policy requires authors to make all data underlying the findings described in their manuscript fully available without restriction, with rare exception. The data should be provided as part of the manuscript or its supporting information, or deposited to a public repository. For example, in addition to summary statistics, the data points behind means, medians and variance measures should be available. If there are restrictions on publicly sharing data—e.g. participant privacy or use of data from a third party—those must be specified.requires authors to make all data underlying the findings described in their manuscript fully available without restriction, with rare exception. The data should be provided as part of the manuscript or its supporting information, or deposited to a public repository. For example, in addition to summary statistics, the data points behind means, medians and variance measures should be available. If there are restrictions on publicly sharing data—e.g. participant privacy or use of data from a third party—those must be specified.

Reviewer #1: Yes

Reviewer #2: Yes

5. Is the manuscript presented in an intelligible fashion and written in standard English?

Reviewer #1: Yes

Reviewer #2: Yes

Reviewer #1: Please ensure the prevalence for severe food insecurity is consistent throughout the abstract, as it is currently listed as both 69.6% and 69.46%.

Could you please clarify the justification for using convenience sampling, as this method inherently introduces selection bias that may limit how well these findings represent the entire camp population?

Please double-check the PTSD prevalence reported in the discussion (3.6%) against the results section (3.0%) to ensure data consistency across the manuscript.

It would be beneficial to expand slightly on why Congolese children showed significantly higher odds for all three mental health outcomes compared to their Burundian peers.

Reviewer #2: Thank you for the opportunity to review this revised manuscript. The authors have thoughtfully and thoroughly addressed the concerns raised in the initial review. The revisions have improved the clarity, rigor, and overall quality of the manuscript. In particular, the authors provided sufficient clarification of the study rationale, methodological decisions, and interpretation of the findings, which strengthens the contribution of this work.

I have no additional substantive concerns, and I believe the manuscript is suitable for publication in its current form. This study makes a meaningful contribution to the literature, and I appreciate the authors’ careful attention to reviewer feedback.

**Do you want your identity to be public for this peer review?** For information about this choice, including consent withdrawal, please see our Privacy Policy..

Reviewer #1: **Yes:** Bala NimmanaBala NimmanaBala NimmanaBala Nimmana

Reviewer #2: No
